# Numerical Investigation of Fracture Behaviour of Polyurethane Adhesives under the Influence of Moisture

**DOI:** 10.3390/polym16182676

**Published:** 2024-09-23

**Authors:** Siva Pavan Josyula, Stefan Diebels

**Affiliations:** Applied Mechanics, Saarland University, 66123 Saarbrücken, Germany; siva.josyula@uni-saarland.de

**Keywords:** polyurethane adhesives, non-linearity, viscoelasticity, ageing, fracture, phase-field method

## Abstract

The mechanical behaviour of polymer adhesives is influenced by the environmental conditions leading to ageing and affecting the integrity of the material. The polymer adhesives have hygroscopic behaviour and tend to absorb moisture from the environment, causing the material to swell without applying external load. The focus of the work is to investigate the viscoelastic material behaviour under ageing conditions. The constitutive equations and the governing equations to numerically investigate the fracture in swollen viscoelastic material are discussed to describe the numerical implementation. Phase-field damage modelling has been used in numerical studies of ductile and brittle materials for a long time. The finite-strain phase-field damage model is used to investigate the fracture behaviour in aged viscoelastic polymer adhesives. The finite-strain viscoelastic model is formulated based on the continuum rheological model by combining spring and Maxwell elements in parallel. Commercially available post-cured crosslinked polyurethane adhesives are used in the current investigation. Post-cured samples of crosslinked polyurethane adhesives are prepared for different humidity conditions under isothermal conditions. These aged samples are used to perform tensile and tear tests and the test data are used to identify the material parameters from the curve fitting process. The experiment and simulation are compared to relate the findings and are the first step forward to improve the method to model crosslinked polymers.

## 1. Introduction

Adhesives are used to bond coated metal and fiber-reinforced plastics (FRPs) in automotive engineering, aerospace applications, shipbuilding, wind energy, and rail vehicle construction. The long-term durability of the bonded joints and the verification process are decisive in manufacturing economically viable bonded structures. Crosslinked network polymer adhesives are becoming increasingly dominant in bonding technology in the manufacturing of lightweight materials. Epoxy and polyurethane adhesives are the most popular chemically crosslinked network polymer adhesives. The present work focuses on polyurethane structural adhesives that are used as structural adhesives in bonding metals, concrete, or polymers in various industries [1]. The higher-stiffness PU adhesives are based on an alcohol–isocyanate chemistry. The curing process forms polyurethane in a primary reaction to produce final mechanical properties. Polyurethane adhesives are used in manufacturing bonded joints, especially in the automotive industry (e.g., BMW I3 life module, BMW M3 bumper beam, Audi, Toyota, Ford, etc.). The interlaminar strength of these adhesives is similar to the FRP adherents, and thus the transfer of forces is optimally utilised reducing frictional resistance at the interface.

These structural adhesives exhibit largely non-linear rate-dependent behaviour due to simultaneous elastic and viscous behaviour. Additionally, polyurethane adhesives are sensitive to the moisture of the environmental conditions due to hygroscopic properties. Hygroscopic behaviour leads to the diffusion of moisture from the environment causing material to age [2]. The ageing of material is classified into chemical and physical ageing. Chemical ageing is an irreversible process due to the chemical interaction between moisture and polymer network resulting in breakage of bonds and forming new bonds, whereas physical ageing is reversible since there are no chemical interactions that influence the physical properties of the material over time. The primary focus of the present work is to formulate a material model to investigate the physical ageing of polyurethane adhesive under the influence of moisture.

The moisture diffusion due to the hygroscopic behaviour leads to the swelling of the polymer without the application of any external loads [3,4]. The swelling of the polymer is purely volumetric leading to the swelling stresses [5,6]. Several theories were proposed and widely investigated to model the swelling behaviour in polymer gels considering chemical and mechanical interactions based on mixing theories [7,8,9,10]. Further, several researchers proposed the modelling of the swelling behaviour under large deformation based on the multiplicative split of the deformation gradient into the swelling and mechanical components based on the finite-strain theory [11,12]. The mechanical deformation gradient describes the viscoelastic behaviour of the material. Material models based on finite-strain viscoelastic mechanical behaviour are modelled primarily for rubber materials [13]. These material models are classified into phenomenological and micro-mechanical models. Phenomenological continuum mechanical models are formulated based on invariants or principal strains [14,15,16,17]. Micro-mechanical models are modelled based on the statistical polymer network theory [18,19,20,21,22]. This article utilises phenomenological formulations to describe the finite-strain viscoelastic behaviour. Herein, a nearly incompressible behaviour is assumed with large deformations, and therefore the deformation is decomposed into volumetric and isochoric parts [23,24]. The decomposition provides a clear description of the physical behaviour of volume and shape-changing parts. Further, the isochoric part is decomposed into elastic and inelastic parts to describe rate-dependent behaviour using continuum-based rheological models.

Modelling of crack propagation is an existing challenge in polymer materials [25,26]. In this context, the crack propagation is well understood within the framework of theoretical continuum mechanics [27]. The energy balance at the crack propagation boundary is described based on Griffith’s criterion. Griffith’s theory states that a crack propagates when the energy release rate at the crack propagation zone is higher than the surface energy built up. The conventional method in modelling cracks separates the material into a broken and intact material by an interface. However, such a method requires a priori knowledge of the exact position of the interface and is complex to model in three-dimensional systems. Therefore, the phase-field method is developed to have a decisive advantage over sharp interface models since the explicit interface tracking becomes redundant [28].

A distinction is made between physical and mechanical approaches in modelling the phase-field material models. The physical model approaches are based on the Ginzburg–Landau phase transformation. In contrast, the mechanical approaches are based on Griffith’s failure theory. A review of the different approaches in modelling phase-field ductile fractures is detailed by Ambati et al. [29]. These models use order parameters to distinguish broken and intact material by minimising the system’s free energy [30]. Phase-field fracture models describe crack propagation in homogeneous materials under different loads [31,32,33], including plastic effects [29,34,35,36] and multi-physics problems [37,38,39]. Based on Griffith’s theory, a model with position-dependent crack resistance was presented by Hossain et al. [40] for studies of fracture strength in materials.

## 2. Material Model Formulation

The constitutive equation and the governing equation are derived within the framework of micro-force balance [41] considering finite-strain theory. The kinematics involved in the non-linearity due to large deformations considering the swelling and nearly incompressible behaviour are discussed in detail. The constitutive and governing equations are derived for the specific choice of free energy functions and discussed in detail within the framework of finite-strain theory. The thermodynamic consistency is discussed in the Appendix A.

### 2.1. Finite-Strain Viscoelasticity

The swelling of the material due to moisture diffusion is incorporated into the mechanical deformation process by multiplicative decomposition of the deformation gradient as
(1)$$\hat{F}=F_{s}\cdot F,$$
where $\hat{F}$, $F_{s}$, and F correspond to the total, swell, and mechanical deformation gradients. The swell deformation gradient is calculated with swell stretch $\lambda_{s}$
(2)$$F_{s}=\lambda_{s}I\;\;\;\mathrm{with}\;\;\;\lambda_{s}={\left(1+mC\right)}^{1/3},$$
where *m* is the slope of the swell stretch for the moisture concentration *C* distributed in the material. The function for swell stretch is formulated to consider the polymer swell due to inhomogeneous moisture distribution. The anomalous moisture diffusion behaviour and the modelling of moisture diffusion are not discussed since the experimental investigations are performed on the saturated samples. The Jacobian of the deformation gradient follows:(3)$$\hat{J}=J_{s}J;\;\mathrm{where}\;\left\{\begin{array}{c} \hat{J}=\text{det}\hat{F}\\ J_{s}=\text{det}\;F_{s}\\ J=\text{det}\;F. \end{array}\right.$$

A nearly incompressible material behaviour motivates the multiplicative decomposition of the deformation gradient tensor into its isochoric and volumetric components. This decomposition of the deformation gradient F is
(4)$$F=F_{\mathrm{vol}}\cdot \bar{F},$$
where $F_{\mathrm{vol}}$ and $\bar{F}$ are the volumetric and isochoric components, respectively. The deformation tensor is decomposed into the elastic $F_{e}$ and inelastic $F_{i}$ parts [23,42,43,44,45,46,47,48]. The decomposition introduces a fictitious intermediate configuration to represent rate-dependent behaviour. Each Maxwell element is introduced with a fictitious intermediate configuration. The decomposition of the deformation gradient is
(5)$$F\;=\;F_{e}\;\cdot \;F_{i}\;.$$

The elastic and inelastic component deformation gradient in Equation (Equation 5) are enforced on the isochoric component of the deformation gradient, thus leading to
(6)$${\bar{F}}_{e}={\left(\text{det}\;F_{e}\right)}^{1/3}F_{e},\;\mathrm{and}\;{\bar{F}}_{i}={\left(\text{det}\;F_{i}\right)}^{1/3}F_{i}.$$

The associated unimodular Cauchy–Green deformation tensors are
(7)$$\begin{array}{cc} \; & {\bar{C}}_{e}=J^{-2/3}{\left({\bar{F}}_{e}\right)}^{T}\cdot {\bar{F}}_{e};\;\;{\bar{C}}_{i}^{j}=J^{-2/3}{\left({\bar{F}}_{i}^{j}\right)}^{T}\cdot {\bar{F}}_{i}^{j}\\ \; & {\bar{B}}_{e}=J^{-2/3}{\left({\bar{F}}_{e}\right)}^{-T}\cdot {\left({\bar{F}}_{e}\right)}^{-1};\;\;{\bar{B}}_{i}^{j}=J^{-2/3}{\left({\bar{F}}_{i}^{j}\right)}^{-T}\cdot {\left({\bar{F}}_{i}^{j}\right)}^{-1}. \end{array}$$

The following relationship applies between the quantities:(8)$${\bar{B}}_{e}=\bar{F}\cdot {\left({\bar{C}}_{i}^{j}\right)}^{-1}\cdot \bar{F}.$$

The time derivative of inelastic Cauchy–Green deformation yields the rate of the inelastic right Cauchy–Green deformation tensor on the reference configuration as follows: (9)$${\dot{\bar{C}}}_{i}^{j}=J^{-2/3}{\left({\left({\bar{F}}_{i}^{j}\right)}^{T}\cdot {\bar{F}}_{i}^{j}\right)}^{.}.$$

The inelastic right Cauchy–Green deformation tensor is regarded as an internal variable in the subsequent continuum mechanical description of the material behaviour. Its rate of inelastic right Cauchy–Green deformation tensor is described by evolution equation [49,50,51]
(10)$$\begin{array}{cc} {\dot{\bar{C}}}_{i}^{j}\; & =\;\frac{4}{r_{j}}\left[\bar{C}-\frac{1}{3}\mathrm{tr}\left(\bar{C}\cdot {\left({\bar{C}}_{i}^{j}\right)}^{-1}\right){\bar{C}}_{i}^{j}\right], \end{array}$$
where $r_{j}$ is the relaxation time associated with individual Maxwell elements. The evolution equation is an outcome of the dissipation inequality described in the Appendix A. The equation is solved with an implicit Euler method in time in combination with a local Newton method in space at each Gaussian point in the framework of the boundary value problem. $I_{1}$ and $I_{3}$ are the first and third invariants of the Cauchy–Green deformation tensor. These invariants are calculated as follows:(11)$$I_{1}=\mathrm{tr}\left(B\right);\;I_{3}=\det\left(C\right)=\det\left(B\right)=J^{2},$$
and the modified counterparts of the invariants are calculated as
(12)$${\bar{I}}_{1}=J^{-2/3}I_{1};\;{\bar{I}}_{3}=\det\left(\bar{C}\right)=\det\left(\bar{B}\right).$$

The viscoelastic material model is based on a continuum mechanical description motivated by a rheological model. Rheological models are described by combining the spring element with the Maxwell element in parallel to model relaxation behaviour. The discrete spectrum of relaxation time is considered by combining the spring element with several Maxwell elements as shown in Figure 1.

A nearly incompressible deformation is assumed in modelling the non-linear viscoelastic behaviour. Therefore, an uncoupled response of free energy [24,52] is used in the formulation. Total mechanical energy of the rheological model consisting of j=1,2,…,n is
(13)$$W_{0}\left(J_{s},J,I_{1}^{\bar{B}},I_{1}^{{\bar{B}}_{e}^{j}}\right)=W_{\mathrm{vol}}\left(J_{s},J\right)+W_{\mathrm{eq}}\left({\bar{I}}_{1}^{\bar{B}}\right)+\sum_{j=1}^{n}W_{\mathrm{neq}}^{j}\left(I_{1}^{{\bar{B}}_{e}^{j}}\right).$$

The equilibrium part of the free energy function is motivated by a polynomial function of the isochoric first invariant of the left Cauchy–Green deformation tensor ${\bar{I}}_{1}^{\bar{B}}$ using the Yeoh model [53,54]
(14)$$W_{\mathrm{eq}}\left({\bar{I}}_{1}^{\bar{B}}\right)=c_{10}\;\left({\bar{I}}_{1}^{\bar{B}}-3\right)+\;c_{20}\;{\left({\bar{I}}_{1}^{\bar{B}}-3\right)}^{2}\;+\;c_{30}\;{\left({\bar{I}}_{1}^{\bar{B}}-3\right)}^{3},$$
where $c_{10},\;c_{20}\;\mathrm{and}\;c_{30}$ are the stiffness parameters. A general quadratic form is considered in current formulation [16,55,56,57,58] of the volumetric part of free energy
(15)$$W_{\mathrm{vol}}\left(J_{s},J\right)=\frac{1}{G}J_{s}{\left(J-1\right)}^{2},$$
where *G* is the compression modulus. The non-equilibrium free energy for j=1,2,…,n Maxwell elements is computed as the sum of the individual energies of Maxwell elements:(16)$$\sum_{j=1}^{n}W_{\mathrm{neq}}^{j}\left(I_{1}^{B_{e}^{j}}\right)=\sum_{j=1}^{n}W_{\mathrm{neq}}^{j}\left({\bar{I}}_{1}^{{\bar{B}}_{e}^{j}}\right)=\sum_{j=1}^{n}c_{10j}\left({\bar{I}}_{1}^{{\bar{B}}_{e}^{j}}-3\right).$$

The corresponding constitutive equation for stresses is calculated as the sum of volumetric, equilibrium, and j=1,2,…,n non-equilibrium stress components:(17)$$T\left(J_{s},J,I_{1}^{\bar{B}},I_{1}^{{\bar{B}}_{e}^{j}}\right)=T_{\mathrm{vol}}\left(J_{s},J\right)+T_{\mathrm{eq}}\left({\bar{I}}_{1}^{\bar{B}}\right)+\sum_{j=1}^{n}T_{\mathrm{neq}}^{j}\left(I_{1}^{{\bar{B}}_{e}^{j}}\right).$$

### 2.2. Phase-Field Damage

This section describes the phase-field damage model used to investigate fracture in the materials that exhibit rate-dependent behaviour due to viscoelastic properties with large deformations. The basic idea behind the variational formulation of the phase-field fracture model is to minimise the free energy by obeying a kinematically admissible displacement field. The free energy *W* is based on Francfort–Marigo functional [59] to describe cracks follows:(18)$$W\left({\bar{I}}_{1}^{\bar{B}},{\bar{I}}_{1}^{{\bar{B}}_{e}^{j}},J,J_{s},\phi \right)=W_{b}\left({\bar{I}}_{1}^{\bar{B}},{\bar{I}}_{1}^{{\bar{B}}_{e}^{j}},J,J_{s},\phi \right)+W_{s}\left(\phi \right),$$
where $W_{b}$ and $W_{s}$ are the bulk and surface energies. The surface energy is given by Griffith’s theory [60] to predict crack initiation and branching:(19)$$W_{s}\left(\phi \right)=\int_{\Gamma }E_{c}d\Gamma .$$

$E_{c}$ is the critical energy release rate to describe the crack resistance of the material. The surface energy is regularised with the crack density functional γ [61] to obtain volume integral
(20)$$W_{s}\left(\phi \right)=\int_{\Omega }E_{c}\gamma \left(\phi ,\mathrm{grad}\;\phi \right)\;\mathrm{dV}.$$

The crack energy density functional introduces phase-field variable $\phi \left(x\right)\in \left[0,1\right]$ to distinguish between intact ϕ(x)=1 and cracked ϕ(x)=0 material. The crack surface density is introduced with a second-order regularised function as
(21)$$\begin{array}{c} \gamma \left(\phi ,\;\mathrm{grad}\;\phi \right)=\left[\frac{1}{2\ell_{f}}{\left(1-\phi \right)}^{2}+\frac{\ell_{f}}{2}{\left|\mathrm{grad}\;\phi \right|}^{2}\right], \end{array}$$
wherein $\ell_{f}$ is the length scale parameter introduced to control the size of the crack zone. The mechanical energy stored in the bulk degrades as the crack propagates with time. A degradation function is introduced to consider the degradation of bulk energy $W_{b}$ as:(22)$$W_{b}\left({\bar{I}}_{1}^{\bar{B}},{\bar{I}}_{1}^{{\bar{B}}_{e}^{j}},J,J_{s},\phi \right)=g\left(\phi \right)W_{0}\left({\bar{I}}_{1}^{\bar{B}},{\bar{I}}_{1}^{{\bar{B}}_{e}^{j}},J,J_{s}\right),$$
where g(ϕ) is the degradation function and $W_{0}$ is the viscoelastic free energy defined in Equation (Equation 13). The degradation function plays a vital role in interpolating stresses to characterise the intact and broken state of the material. The degradation of bulk energy for intact and broken state defined by phase-field variable $\phi \in \left[0,\;1\right]$ have to satisfy conditions
(23)$$\begin{array}{cc} \; & W_{b}\left({\bar{I}}_{1}^{\bar{B}},{\bar{I}}_{1}^{{\bar{B}}_{e}^{j}},J,J_{s},\phi =1\right)=W_{0}\left({\bar{I}}_{1}^{\bar{B}},{\bar{I}}_{1}^{{\bar{B}}_{e}^{j}},J,J_{s}\right),\;\;W_{b}\left({\bar{I}}_{1}^{\bar{B}},{\bar{I}}_{1}^{{\bar{B}}_{e}^{j}},J,J_{s},\phi =0\right)=0\;,\\ \; & \partial W_{b}\left({\bar{I}}_{1}^{\bar{B}},{\bar{I}}_{1}^{{\bar{B}}_{e}^{j}},J,J_{s},\phi \right)<1\;\;\;\mathrm{and}\;\;\;\partial W_{b}\left({\bar{I}}_{1}^{\bar{B}},{\bar{I}}_{1}^{{\bar{B}}_{e}^{j}},J,J_{s},\phi =0\right)=0. \end{array}$$

Equations (Equation 20)–(Equation 22) are substituted in the Equation (Equation 18) and integrated over the volume to derive the free energy density [31,59]
(24)$$\begin{array}{cc} W\left({\bar{I}}_{1}^{\bar{B}},{\bar{I}}_{1}^{{\bar{B}}_{e}^{j}},J,J_{s},\phi \right)= & \int_{\Omega }g\left(\phi \right)W_{0}\left({\bar{I}}_{1}^{\bar{B}},{\bar{I}}_{1}^{{\bar{B}}_{e}^{j}},J,J_{s}\right)\mathrm{dV}\;\;+\\ \; & \int_{\Omega }E_{c}\left[\frac{1}{2\ell_{f}}{\left(1-\phi \right)}^{2}+\frac{\ell_{f}}{2}{\left|\mathrm{grad}\;\phi \right|}^{2}\right]\mathrm{dV}. \end{array}$$

In this article, a monotonically decreasing function is considered to describe the decay of the bulk energy. A second-order degradation function is considered with an additional regularisation parameter ζ [61,62,63] to interpolate the bulk energy
(25)$$g\left(\phi \right)=\left(1-\zeta \right)\phi^{2}+\zeta .$$

The regularisation parameter ζ>0 is employed to guarantee a converged solution. A smaller value is selected to avoid overestimation of mechanical energy [61,64,65]. The energy degradation function satisfies the condition
(26)$$g\left(\phi =0\right)=0,\;g\left(\phi =1\right)=1\;\mathrm{and}\;g^{\prime }\left(\phi =0\right)=0,$$
where g(ϕ=0)=0 damaged material, g(ϕ=1)=1 describes the intact material and $g^{\prime }\left(\phi =0\right)=0$ controls the stored mechanical energy in the phase-field evolution equation.

## 3. Governing Balance Equations

The weak form of the free energy function is derived by applying the variational principle to the total potential energy with the field variables $\left(u,\phi \right)$
(27)$$\delta W=\left(\frac{\partial W}{\partial u}\right):\delta u+\left(\frac{\partial W}{\partial \phi }\right):\delta \phi .$$

Furthermore, the continuum domain Ω is integrated over the total volume dV leading to the weak form for the admissible test functions of phase-field δϕ and displacement field δu
(28)$$\begin{array}{cc} \delta W=\int_{\Omega } & \left\{g\left(\phi \right)\;T:{\mathrm{grad}}^{s}\delta u+g^{\prime }\left(\phi \right)\;\delta \phi W_{0}\right\}\mathrm{dV}+\\ \; & \int_{\Omega }\left\{E_{c}\left[-\frac{1}{\ell_{f}}\left(1-\phi \right)\delta \phi +\ell_{f}\;\mathrm{grad}\;\phi \;\mathrm{grad}\delta \phi \right]\right\}\mathrm{dV}, \end{array}$$
${\mathrm{grad}}^{s}\delta u=\frac{1}{2}\left[\mathrm{grad}\delta u+{\left(\mathrm{grad}\delta u\right)}^{T}\right]$ is involved due to symmetric stress tensor. After substituting the degradation function defined in Equation (Equation 25) and the derivative of degradation function $g^{\prime }\left(\phi \right)=\partial g(\phi )/\partial \phi$ in the coupled form given in Equation (Equation 28) follows:(29)$$\begin{array}{cc} \delta W=\int_{\Omega } & \left\{\left[\left(1-\zeta \right)\phi^{2}+\zeta \right]T:{\mathrm{grad}}^{s}\delta u\right\}\mathrm{dV}+\\ \int_{\Omega } & \left\{2\left(1-\zeta \right)\phi \;\delta \phi W_{0}+E_{c}\left[-\frac{1}{\ell_{f}}\left(1-\phi \right)\delta \phi +\ell_{f}\;\mathrm{grad}\;\phi \;\mathrm{grad}\delta \phi \right]\right\}\mathrm{dV}. \end{array}$$

The strong form of the coupled formulation gives the local statement for the phase-field method and is derived by applying the divergence principle
(30)$$\begin{array}{cc} \mathrm{div}\left(\left[\left(1-\zeta \right)\phi^{2}+\zeta \right]T\right) & =0\\ \underset{\mathrm{driving}\;\mathrm{force}}{\underset{︸}{2\left(1-\zeta \right)\phi \;W_{0}}}+\underset{\mathrm{resistance}\;\mathrm{to}\;\mathrm{crack}}{\underset{︸}{E_{c}\left[-\frac{1}{\ell_{f}}\left(1-\phi \right)+\ell_{f}\;\mathrm{div}\phi \right]}} & =0. \end{array}$$

The first equation in (Equation 30) corresponds to the balance of momentum, while the latter equation describes the phase-field evolution of the diffusive crack. The first term of the phase-field evolution is responsible for driving the crack, and the second term refers to the geometric resistance to the propagation of the crack. $W_{0}$ is the energy stored in material domain with $W_{0}=\max_{0<\phi <t}W_{0}\left(x,\phi \right)$ to avoid an irreversibility in the crack propagation [37].

### 3.1. Finite Element Implementation

It is convenient to express the partial differential Equations (Equation 30) in their weak forms to develop a numerical solution scheme using finite element method:(31)$$\begin{array}{cc} r_{i}^{u}=\int_{\Omega }\left\{\left[\left(1-\zeta \right)\phi^{2}+\zeta \right]T:{\mathrm{grad}}^{s}\delta u\right\}\mathrm{dV} & =0\\ r_{i}^{\phi }=\;\;\int_{\Omega }\;\;\left\{2\left(1-\zeta \right)\phi \;\delta \phi W_{0}\;+\;E_{c}\left[-\frac{1}{\ell_{f}}\left(1-\phi \right)\delta \phi \;+\;\ell_{f}\;\mathrm{grad}\phi \;\mathrm{grad}\delta \phi \right]\right\}\mathrm{dV} & =0. \end{array}$$

In this context, the displacement u and phase-field variable ϕ are discretised in space as
(32)$$u=\sum_{i=1}^{n_{\mathrm{ele}}}N_{i}^{u}u_{i}\;\phi =\sum_{i=1}^{n_{\mathrm{ele}}}N_{i}^{\phi }\phi_{i},$$
where $N_{i}^{\phi }$ is the shape function concerning the phase-field variable and $N_{i}^{u}$ is the displacement shape function used to interpolate between the quantities at the quadrature points. The displacement shape function in three dimensions is given by:(33)$$N_{i}^{u}=\left[\begin{array}{ccc} N_{i} & 0 & 0\\ 0 & N_{i} & 0\\ 0 & 0 & N_{i} \end{array}\right].$$

In Equation (Equation 33), $N_{i}$ is the value of the shape function of the displacement field $u_{i}\;=\;{\left(u_{x},u_{y},u_{z}\right)}^{T}$ at the quadrature points associated with the respective nodes. The gradient of the phase-field variable $\phi_{i}$ follows: (34)$$\mathrm{grad}\phi =\sum_{i=1}^{n_{\mathrm{ele}}}S_{i}^{\phi }\phi_{i}.$$

Herein, the S matrix is introduced as
(35)$$S_{i}^{u}=\left[\begin{array}{ccc} N_{i,x} & 0 & 0\\ 0 & N_{i,y} & 0\\ 0 & 0 & N_{i,z}\\ N_{i,y} & N_{i,x} & 0\\ 0 & N_{i,z} & N_{i,y}\\ N_{i,z} & 0 & N_{i,x} \end{array}\right]\;S_{i}^{\phi }=\left[\begin{array}{c} N_{i,x}\\ N_{i,y}\\ N_{i,z} \end{array}\right],$$
where $N_{i,x}$, $N_{i,y}$ and $N_{i,z}$ are the derivatives of the shape functions evaluated as $\partial N_{i}/\partial x$, $\partial N_{i}/\partial y$ and $\partial N_{i}/\partial z$. In the same way, virtual quantities of the displacement and phase-field variables are approximated as
(36)$$\begin{array}{c} \delta u=\sum_{i=1}^{n_{\mathrm{ele}}}N_{i}^{u}\delta u_{i}\;\delta \phi =\sum_{i=1}^{n_{\mathrm{ele}}}N_{i}^{\phi }\delta \phi_{i}\\ {\mathrm{grad}}^{s}\delta u=\sum_{i=1}^{n_{\mathrm{ele}}}S_{i}^{\delta u}u_{i}\;\mathrm{grad}\;\delta \phi =\sum_{i=1}^{n_{\mathrm{ele}}}S_{i}^{\phi }{\delta \phi }_{i}. \end{array}$$

The coupled system of equations is non-linear, and therefore the coupled problem is solved iteratively using the Newton–Raphson method. The finite element formulation to solve the coupled system of equations with an incremental method follows:(37)$$\left[\begin{array}{cc} K^{\mathrm{uu}} & K^{u\phi }\\ K^{\phi u} & K^{\phi \phi } \end{array}\right]\left[\begin{array}{c} du\\ d\phi \end{array}\right]=\left[\begin{array}{c} -r_{u}\left(u_{i}\right)\\ -r_{\phi }\left(\phi_{i}\right) \end{array}\right].$$

Since the primary variables defined in Equation (Equation 32) hold for the arbitrary values δu and δϕ, the residuals of the coupled system of equations defined in Equation (Equation 31) are expressed in term of the virtual quantities given with Equation (Equation 36) as
(38)$$\begin{array}{cc} r_{i}^{u}=\int_{\Omega }\left\{\left[\left(1-\zeta \right)\phi^{2}+\zeta \right]T:{\left(S_{i}^{u}\right)}^{T}\right\}\mathrm{dV} & =0,\\ r_{i}^{\phi }=\int_{\Omega }\left\{2\left(1-\zeta \right)\phi \;N_{i}W_{0}+E_{c}\left[-\frac{1}{\ell_{f}}\left(1-\phi \right)N_{i}+\ell_{f}{\left(S_{i}^{\phi }\right)}^{T}S_{j}^{\phi }\right]\right\}\mathrm{dV} & =0, \end{array}$$
and the elements of the tangent matrix are
(39)$$\begin{array}{cc} \; & K_{i,j}^{\mathrm{uu}}=\frac{\partial r_{i}^{u}}{\partial u_{j}}=\int_{\Omega }\left(\left(1-\zeta \right)\phi^{2}+\zeta \right)\left(S_{i}^{u}:\overset{4}{\kappa }:S_{j}^{u}+S_{i}^{u}:\left[T\cdot S_{j}^{u}\right]\right)\mathrm{dV},\\ \; & K_{i,j}^{u\phi }=\frac{\partial r_{i}^{u}}{\partial \phi_{j}}=\int_{\Omega }2\left(1-\zeta \right)\;\phi \;S_{i}^{u}:T^{T}N_{j}^{u}\;\mathrm{dV},\\ \; & K_{i,j}^{\phi u}=\frac{\partial r_{i}^{\phi }}{\partial u_{j}}=\int_{\Omega }2\left(1-\zeta \right)\;\phi \;N_{i}^{u}\;T^{T}:S_{j}^{u}\;\mathrm{dV},\\ \; & K_{i,j}^{\phi \phi }=\frac{\partial r_{i}^{\phi }}{\partial \phi_{j}}=\int_{\Omega }\left\{\left(1-\zeta \right)W_{0}\;N_{i}^{\phi }\;N_{j}^{\phi }+E_{c}\left[\frac{1}{\ell_{f}}N_{i}^{\phi }\;N_{j}^{\phi }+\ell_{f}{\left(S_{i}^{\phi }\right)}^{T}:S_{j}^{\phi }\right]\right\}\mathrm{dV}. \end{array}$$

The coupled system of equations is solved simultaneously using a monolithic approach with Newton’s iterative method.

### 3.2. Boundary Conditions

The boundary conditions are postulated for the displacement field variable u and the phase-field damage variable ϕ to solve the phase-field damage formulation. To this end, the surface ∂Ω is decomposed to the primary fields, the displacement and damage fields
(40)$$\partial \Omega =\partial \Omega_{u}^{D}\cup \partial \Omega_{t}^{N}\;\;\mathrm{and}\;\;\partial \Omega =\partial \Omega_{\phi }^{D}\cup \partial \Omega_{\nabla \phi }^{N}$$
with $\partial \Omega_{u}^{D}\cap \partial \Omega_{u}^{N}=⌀$ and $\partial \Omega_{\phi }^{D}\cap \partial \Omega_{\nabla \phi }^{N}=⌀$. The prescribed displacement u and traction t of the mechanical problem on the boundaries are postulated with the Dirichlet and Neumann boundary conditions
(41)$$u\left(x,t\right)=u_{D}\left(x,t\right)\;\;\mathrm{on}\;\;\partial \Omega_{u}^{D}\;\;\mathrm{and}\;\;T\cdot n=t\;\;\mathrm{on}\;\;\partial \Omega_{t}^{N}.$$

For the phase-field damage, the cracked region is constrained by the Dirichlet and the Neumann boundary conditions on the crack surface with
(42)$$\phi \left(x,t\right)=0\;\mathrm{at}\;x\in \partial \Omega_{\phi }^{D}\;\;\mathrm{and}\;\;\nabla \phi \cdot n=0\;\;\mathrm{on}\;\;\partial \Omega_{\nabla \phi }^{N}.$$

## 4. Results

In this work, the coupled material model is developed to investigate damage behaviour in crosslinked polyurethane adhesives. The chemical structure of basic crosslinked polyurethane adhesive is manufactured by mixing two polyether alcohol components (resin) and an isocyanate component (hardener) without additives [2,66]. The moisture influence on the mechanical behaviour is investigated on the industrially available polyurethane adhesive manufactured by Dupont under the trade name Betaforce 2850L. The experimental investigation performed on the commercial polyurethane adhesive Betaforce 2850L, called adhesive-A in project IGF-project 19730 N [67], is used in the present work for numerical investigation. The tensile tests performed on the aged samples are used to identify the viscoelastic parameters. The aged samples are prepared at four different relative humidity conditions 0%r.H, 29%r.H, 67%r.H, and 100%r.H at an isothermal condition of 60 °C. The material reaches to the saturation states in 1hr as observed in the moisture diffusion tests [67]. In the current study, the swelling stretch is assumed to be negligible as the moisture diffusion time is small to reach the saturation state.

The viscoelastic model is formulated by using a spring element connected in parallel to the four Maxwell elements. The relaxation times of the Maxwell elements are assumed constant irrespective of ageing conditions. The assumption of constant relaxation time is analogous to the viscoelastic behaviour proposed in [68,69] to fit the loading rates used in the experimental investigation. The stiffness parameters of the polyurethane adhesive were identified with the curve fitting process using Matlab optimisation toolbox. The optimisation algorithms are classified into gradient-free and gradient methods. The solution from gradient methods may not be unique due to local minima and depends on the start value. Therefore, a gradient-free algorithm proposed by Nelder and Mead [70] is used to identify stiffness parameters by curve fitting. The identified parameters of the viscoelastic material for all the investigated aged samples are listed in Table 1.

The aged samples are prepared for different humidity conditions. The aged samples are investigated for tensile behaviour under $0.0005\;s^{-1}$ strain rate at an isothermal condition of 60°C. The experimental and simulation data are compared in Figure 2. The deviation between the simulation and test results is agreeably small to use in the further investigation of damage behaviour.

Further, the experimental investigation is conducted on the post-curved aged sample at a higher strain rate of $0.05\;s^{-1}$. The viscoelastic parameters listed in Table 1 are used in the numerical simulation with a higher strain rate. The numerical simulation and test results are compared together to investigate the sensitivity of the identified material parameters. Figure 3 shows the comparison of the simulation and test results. The simulation results show a larger deviation for the maximum stretch due to the large influence of the negative stiffness parameter defined for $c_{20}$ to capture the S-structure.

The focus of this work is to understand the damage behaviour in the polyurethane adhesive due to tensile load for samples aged under the influence of moisture. To this end, the sample preparation and experimental investigation are carried out by following DIN ISO 34-1. Angular specimens proposed in DIN ISO 34-1 (shown in Figure 4) are prepared with a thickness of 2mm to perform the tear test.

It is a well-established fact that the numerical solution obtained from the finite element method depends on the size of the finite element mesh, especially in the case of damage evolution. The crack initiates and propagates at the cross-section of the notch until failure. The mesh at the region of crack is refined to resolve to capture steep gradients at the region of crack propagation. The mesh is refined at the transition zone satisfying the mesh refinement condition proposed by Miehe et al. [33]. According to the proposed criteria, the initial crack length $\ell_{f}\;\mathrm{mm}$ and the mesh size hmm follow:(43)$$h\ll \frac{\ell_{f}}{2}.$$

The phase-field damage model requires a finite element mesh to resolve the damage evolution and produce better numerical results; however, the solution with the refined mesh is computationally expensive. Therefore, a mesh sensitivity study is performed on the V-shaped notch specimen analogous to the angular specimen shown in Figure 4. The V-shaped notch specimen is applied with the tensile load boundary conditions (see Figure 5).

The mesh sensitivity study is performed on the dry sample viscoelastic parameters listed in Table 1 and a critical energy release rate $E_{c}=4.18\;N/\mathrm{mm}$. An initial crack of 9.31mm is used in the experimental investigations of damage behaviour on the post-cured polyurethane adhesives. Therefore, a length-scale parameter $\ell_{f}=9.31\;\mathrm{mm}$ is used in the mesh sensitivity study in the damage investigation. The stress concentration is maximum at the region of the notch. As a result, the crack initiates and propagates at the notch. Therefore, the sample is discretised with a locally refined mesh in the region of the notch. The mesh sensitivity study is investigated for six different mesh sizes hmm to understand the sensitivity of the mesh in crack propagation.

The load-displacement curves of the different mesh sizes are shown in Figure 6. The initial slope of the load-displacement curves is approximately the same in the mesh sensitivity study comparison. However, these curves deviate to a maximum of ≤15% at the peak force of failure due to the mesh size. The deviation is due to the approximation of the steep gradients at the locally refined mesh. In the current investigation of failure in polyurethane adhesive, a mesh size of h=1.23mm is adopted to compromise with the accuracy of the numerical results and reduce computational effort.

Aged samples are manufactured in different humidity conditions at an isothermal condition of 60°C to investigate the influence of moisture on tear strength. Similar to tensile test samples, the angular specimens are aged under humid atmospheric conditions under 0%r.H, 29%r.H, 67%r.H, and 100%r.H conditions at 60 °C. To perform a tear test, an initial crack of 9.13mm is imposed on the samples at the notch. Material is clamped approximately at 22mm on both free ends, which are assumed to be rigid. Therefore, the clamped volume of material is not considered in the numerical investigation; see Figure 7.

The crack length-scale parameter is set to the length of the crack imparted in the sample as $\ell_{f}=9.31\;\mathrm{mm}$ and the length of mesh size of h=1.23mm is maintained to satisfy the mesh refinement condition. Finally, displacement boundary condition $u_{y}=0.011\;\mathrm{mm}$ is applied to investigate the failure. The finite element model is solved monolithically using Newton’s iterative method in a quasi-static process. The solution scheme is solved in several time steps with a constant time increment until fracture. The spatially discretised model is defined with the viscoelastic parameters listed in Table 1. The critical energy release rate is an essential parameter required for the phase-field material model. This parameter is identified from the fitting force-displacement curve from numerical analysis with the tear test using Matlab optimization toolbox [70]. The optimal critical energy release rates $E_{c}$ listed in Table 2 are identified individually for the dry (0%r.H) and aged samples (humidity conditions: 29%r.H., 67%r.H. and 100%r.H.) manufactured under the isothermal condition of 60 °C.

The critical energy release rate of the aged samples shown in Figure 8 indicates that the adhesive material becomes ductile under the influence of moisture, thus leading to an increase in the critical energy release rate. The critical energy release rate is minimum for the dry sample and reaches a maximum for the sample saturated at 29% relative humidity.

The force-displacement data from the simulation are compared with the experimental test data to validate the identified optimal material parameters. The test used in the comparison is the mean value calculated from the series of three angular test samples manufactured at the same atmospheric conditions [67].

Figure 9 compares the experimental and the simulation data, where the deviation between the curves is calculated using the root-mean-square method (RMS) and plotted with the error bars. The deviation in the form of small error bars indicates the problem is well posed, thus validating the identified material parameters. The deviation between the experimental and simulation is due to the compromise in the mesh refinement employed to compromise with the simulation effort. The step gradients of the phase-field damage variable ϕ at the region of crack propagation do not converge to the experimental data due to a compromise in the mesh size. A higher-order phase-field damage model is needed for accurate and faster numerical analysis, thus resolving the numerical deviation.

## 5. Conclusions and Future Work

In this article, viscoelastic behaviour is modelled using on the rheological model based on continuum mechanics by combining elastic spring and Maxwell elements. Four Maxwell elements are used in combination with elastic spring elements in parallel and the energies are defined using phenomenological models. The viscoelastic model is coupled with the phase-field model to investigate the damage behaviour of the aged polyurethane adhesives under the influence of moisture. The phase-field damage model is a promising damage material model based on the Griffith fracture energy. These fracture models describe the crack evolution with a continuous order variable differentiating the intact and damaged material with diffusive cracks.

The experimental investigations performed on a commercial polyurethane adhesive called adhesive-A in IGF-project 19730 N are used for numerical investigation. The fracture behaviour of the material is investigated under the influence of moisture at an isothermal condition. Aged samples are prepared for 29% r.H., 67% r.H. and 100% r.H. conditions at 60 °C. Viscoelastic material model parameters are identified from the tensile test performed on the dry and aged samples. These viscoelastic parameters are used in the numerical investigation of the damage behaviour in the aged samples using the phase-field damage model. The tear strength on the samples is performed based on the DIN ISO 34-1 standard using angular specimens. The critical energy release rate is identified using an optimization algorithm with Matlab toolbox. The deviation between the experimental and simulation results is negligible validating the material parameters.

The proposed material model considers the influence of the swelling due to moisture diffusion on mechanical behaviour. In the present study, the swelling stretch due to moisture diffusion is assumed to be negligible due to the fact that the cross-linked polyurethane adhesives tend to reach the saturation state in a short period of time, as observed in the gravimetric tests to investigate diffusion behaviour in IGF-project 19730 N. A sensitivity study is conducted to investigate the influence of the swelling stretch on the fracture behaviour with $1<\lambda_{s}<1.7$ for the sample saturated at the 100% r.H condition. Swelling stretch shows a significant increase in the total stresses with an increase in the swell stretch (see Figure 10). This consideration of the swell stretch leads to the decrease in the viscoelastic stiffness parameters identified from fitting experimental results.

The motivation of the current topic is to understand the influence of moisture on the mechanical behaviour investigated on saturated samples. The finite-strain phase-field damage model needs to be coupled with the diffusion model to understand mechanical behaviour and swelling influence for the unsaturated materials with inhomogeneous moisture distribution. The crosslinked polyurethane adhesives are formed by a network of chains consisting of shorter and longer chain distribution. The chain distribution shows softening behaviour under the loading conditions since the shorter chains become inactive in sharing the load with longer chains with the increase in the loads. The proposed material model based on the phenomenological approach does not consider the chain distribution, and the stiffness parameters are identified from the curve fitting process with no explanation of physical behaviour. Therefore, it is necessary to model the mechanical behaviour using a micro-mechanical model based on the polymer chain statistics.

The present investigation is performed and investigated with aged samples under an isothermal condition. Therefore, the coupled material model does not account for the softening of material due to the temperature material parameters. The present work also assumes the swelling stretch due to the ambient temperature is negligible in comparison to the swell deformation due to moisture diffusion. The moisture diffusion is influenced by the temperature; therefore, the ageing of the material has to be investigated for different temperatures. Experimental investigations are needed to understand the swell stretch due to moisture diffusion and ambient temperature. Further, the material parameters required to investigate the swell behaviour must be identified and validated to numerically investigate the mechanical behaviour with inhomogeneous moisture distribution.

## Figures and Tables

**Figure 1 polymers-16-02676-f001:**
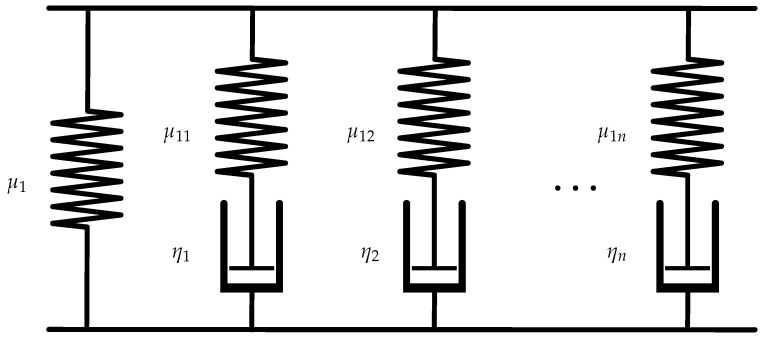
Rheological model of the viscoelasticity with *n* Maxwell elements.

**Figure 2 polymers-16-02676-f002:**
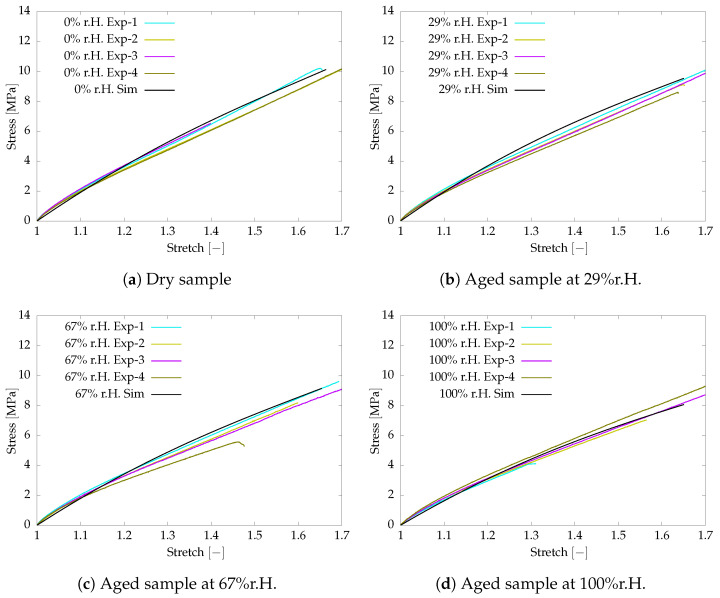
Tensile test and simulation data comparison of aged samples at smaller strain rate $0.0005\;s^{-1}$.

**Figure 3 polymers-16-02676-f003:**
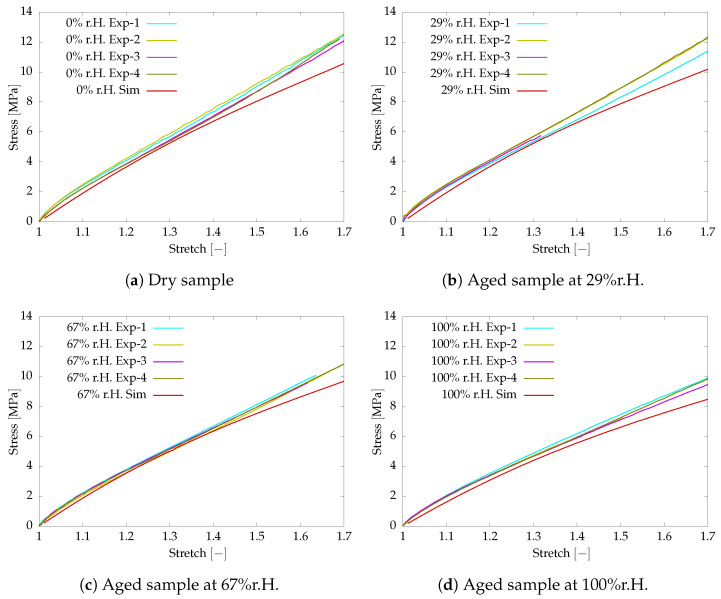
Tensile test and simulation data comparison of aged samples at larger strain rate $0.05\;s^{-1}$.

**Figure 4 polymers-16-02676-f004:**
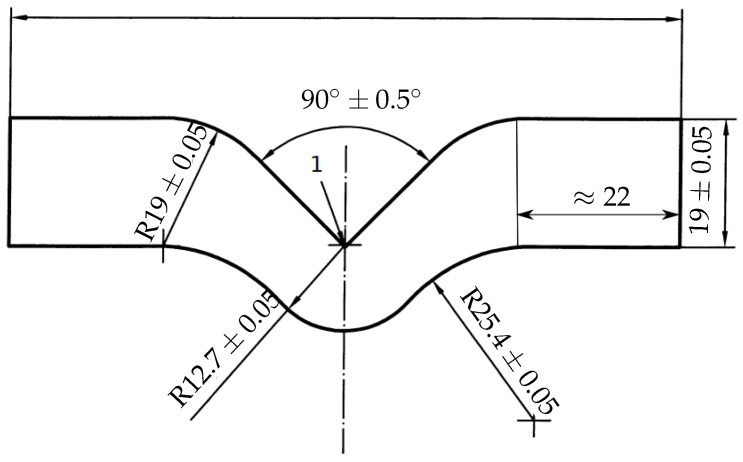
Geometry of the angle sample based on DIN ISO 34-1: all dimensions are in millimetres.

**Figure 5 polymers-16-02676-f005:**
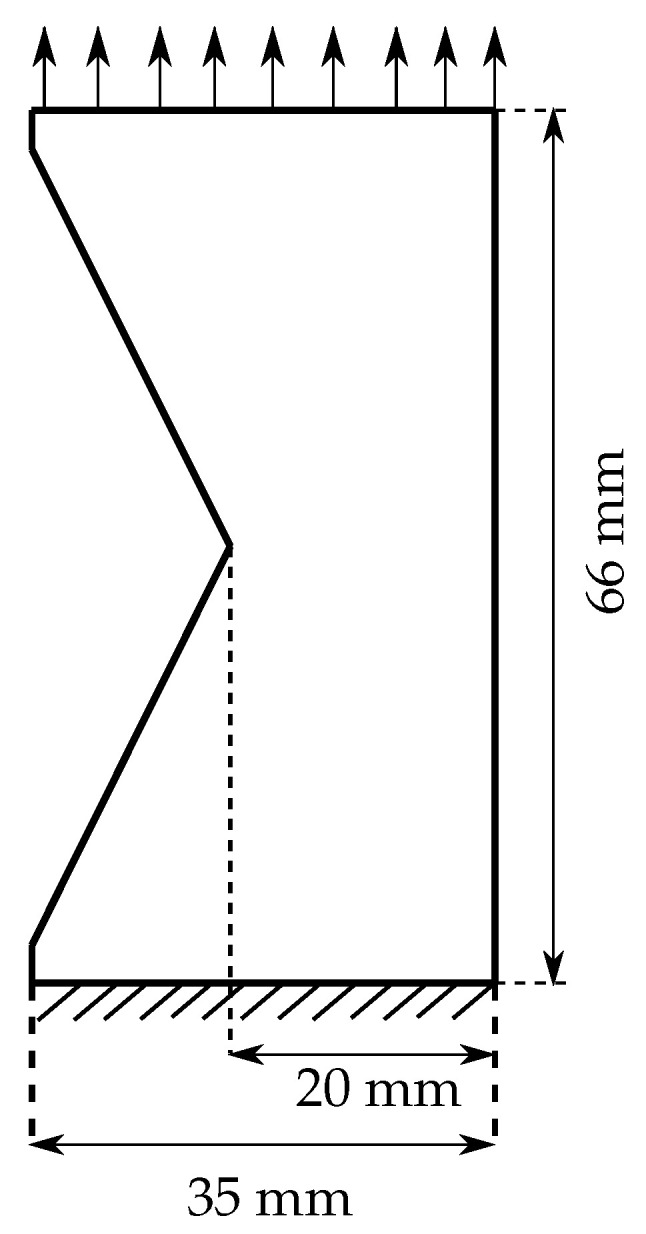
V-shaped notch sample of thickness 2mm applied with tensile boundary conditions.

**Figure 6 polymers-16-02676-f006:**
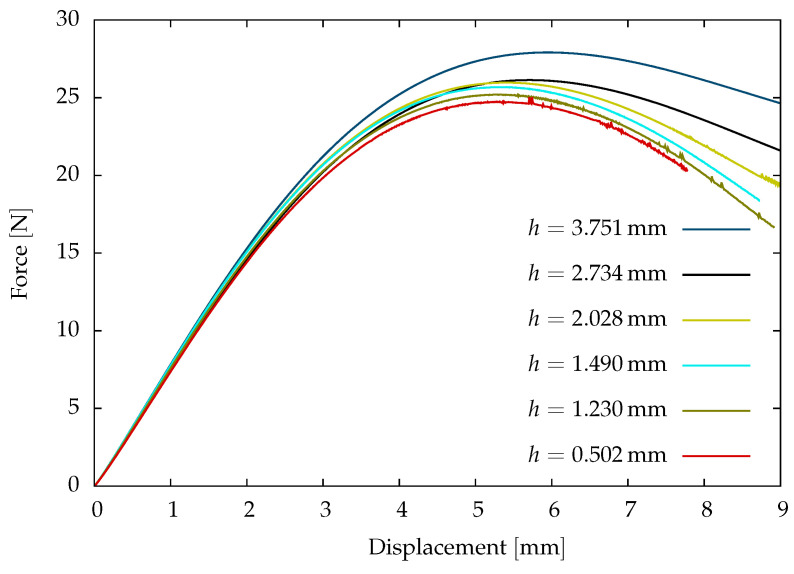
Computational results of single-edge V-shaped notch sample with different mesh length.

**Figure 7 polymers-16-02676-f007:**
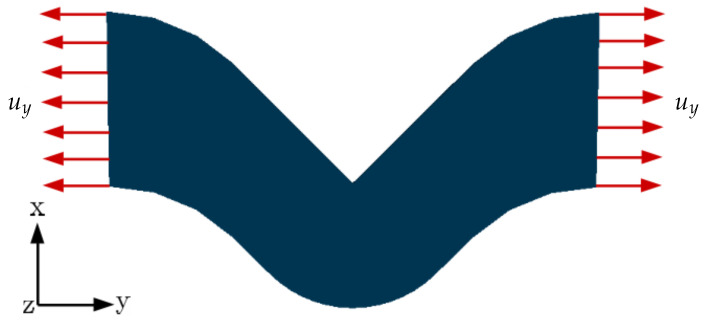
Model is applied with the displacement boundary conditions as tensile loading until failure.

**Figure 8 polymers-16-02676-f008:**
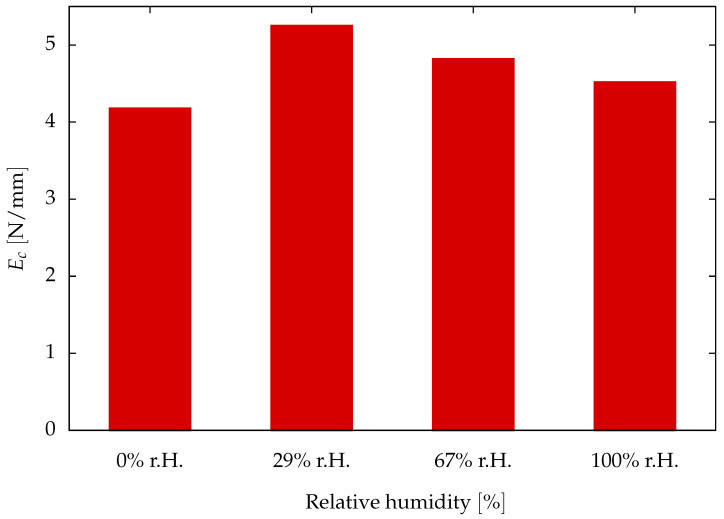
Critical fracture energy release rate $E_{c}$ of adhesive-A samples aged at different relative humidities (r.H.) in the atmosphere at 60°C.

**Figure 9 polymers-16-02676-f009:**
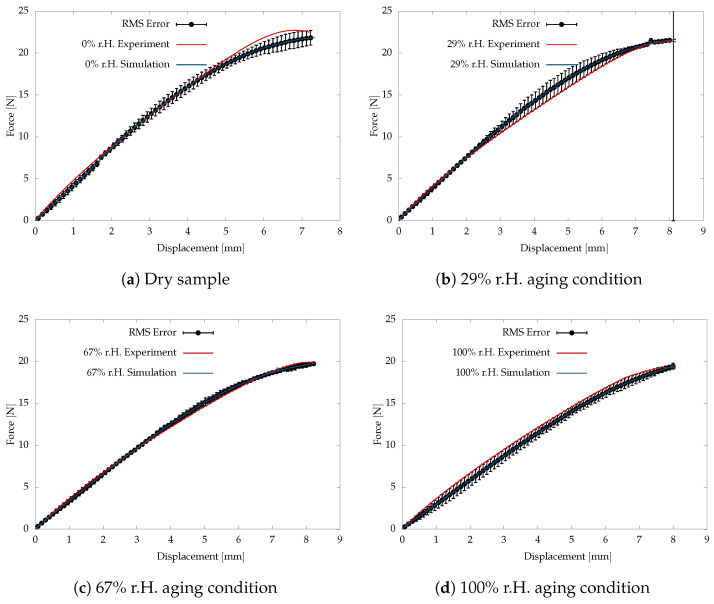
Tear tests are performed on angular samples aged under different humidity conditions at 60°C.

**Figure 10 polymers-16-02676-f010:**
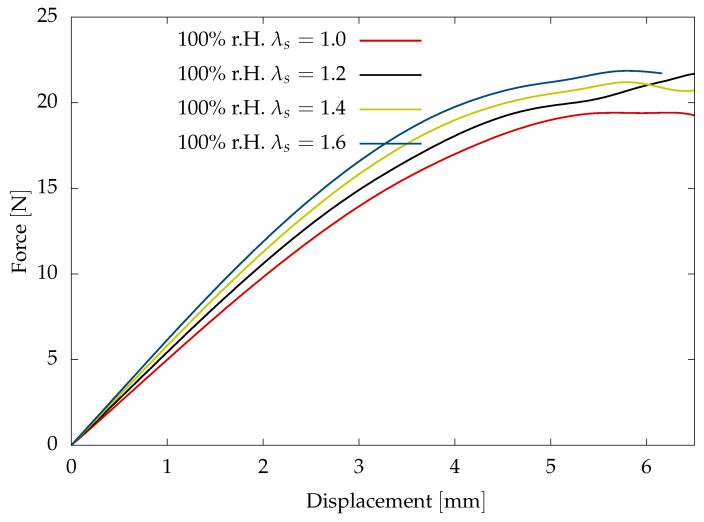
Sensitivity study to investigate the influence of the swell stretch on the fracture behaviour.

**Table 1 polymers-16-02676-t001:** Materialparameters of the finite-strain viscoelastic material model identified for the different relative humidity atmospheres at 60 °C.

Material Parameters of Finite-Strain Viscoelastic Model						
		Relaxation Times [s]	0% r.H.	29% r.H.	67% r.H.	100% r.H.
Equilibrium	$c_{10}\;\left[\mathrm{MPa}\right]$		9.886	7.886	7.196	7.072
	$c_{20}\;\left[\mathrm{MPa}\right]$		−1.414	−1.357	−1.122	−1.128
	$c_{30}\;\left[\mathrm{MPa}\right]$		3.214	1.443	0.918	0.872
	D[MPa]		0.306	0.244	0.241	0.314
Non-equilibrium	$c_{101}\;\left[\mathrm{MPa}\right]$	0.5	4.886	2.886	2.296	2.172
	$c_{102}\;\left[\mathrm{MPa}\right]$	10	0.886	0.231	0.139	0.107
	$c_{103}\;\left[\mathrm{MPa}\right]$	100	0.055	0.017	0.014	0.011
	$c_{104}\;\left[\mathrm{MPa}\right]$	1000	0.005	0.003	0.002	0.001

**Table 2 polymers-16-02676-t002:** Critical energy release rate identified for dry and aged samples prepared and tested at an isothermal condition of 60 °C.

Identified Critical Energy Release Rate				
ageing condition	0% r.H.	29% r.H.	67% r.H.	100% r.H.
$E_{c}\;\left[N/\mathrm{mm}\right]$	4.18N/mm	5.25N/mm	4.82N/mm	4.52N/mm

## Data Availability

The raw data supporting the conclusions of this article will be made available by the authors on request.

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
