# Peer review of "Numerical Investigation of Fracture Behaviour of Polyurethane Adhesives under the Influence of Moisture"

_polymers, 2024, doi:10.3390/polym16182676_

Round 1
Reviewer 1 Report
Comments and Suggestions for Authors
The manuscript “Numerical investigation of fracture behaviour of polyurethane adhesives under the influence of moisture” describes a finite-strain viscoelastic material model coupled with the phase-field model to investigate the fracture behaviour of these adhesives in different moisture.
The results were compared, and the conclusions were made.
The manuscript is original and has merit, but it needs to be improved.
What is the technological/scientific contribution of this paper?
The chemical properties of the adhesive need to be presented.
What is the chemical structure of the adhesive?
Does the adhesive need a curing agent to harden?
it is necessary to write more text on the polyurethane adhesive. Taking this study into account, where will it be used?
The study did not consider the crystallization process, shrinkage process or swelling process in the model. The adhesive behaviour can be influenced by be all these process when subjected to the temperature and the moisture conditions.
The authors must explain in detail the strengths of this work and comparing with those found in the literature.
https://doi.org/10.1177/00219983211012
In the Conclusion section:
Emphasize the main novelty and advantages of the results obtained.
Provide any further outline to proceed with the study in the future.
References are generally not cited in the results.
Author Response
Comment 1: What is the technological/scientific contribution of this paper?
Response 1: Here we discuss the phase-field damage model coupled with viscoelastic behaviour under large deformations. The swelling stretch is considered in the coupled formulation, although the assumption of the negligible swell stretch.
Comment 2: The chemical properties of the adhesive need to be presented.
Response 2: The structure of the crosslinked polyurethane adhesive is discussed with a basic PU9010 which is a chemically crosslinking base polymer without additives, composed of two aliphatic alcohol components (resin) and an aromatic isocyanate component (hardener). The experimental investigation is performed on the commercial polyurethane adhesives. The chemical properties and structure is unknown.
Comment 3: Taking this study into account, where will it be used?
Response 3: Crosslinked polyurethane adhesive is a structural adhesives and they are mainly used in the manufacturing of the automotive lightweight structures.
Comment 4: The study did not consider the crystallization process, shrinkage process or swelling process in the model. The adhesive behaviour can be influenced by be all these process when subjected to the temperature and the moisture conditions.
Response 4: The mechanical behavuiour is investigated on the post-cured samples. Hence the crystallization and shrinkage or swelling process is considered. Swelling due to moisture diffusion is considered in the material model.
Comment 5: The authors must explain in detail the strengths of this work and comparing with those found in the literature.
https://doi.org/10.1177/00219983211012
Response 5: Sorry, I am not able to access the literature provided above. If doi refers to my submission to polymers, I have revised submission detailing the scope and results.
Comment 6: In the Conclusion section: Emphasize the main novelty and advantages of the results obtained. Provide any further outline to proceed.
Response 6: Future prospects of the study is discussed in the section “conclusions and future work”
Reviewer 2 Report
Comments and Suggestions for Authors
It is suggested to mention the related references for the given data in the caption of Table 1, as is shown in the text.
Please describe the mesh dependency method in detail. Which parameter was considered for the mesh size analysis?
The root-mean-square (RMS) of the error should be calculated for the given results in Fig. 3, and the authors should discuss the origin of the errors.
The main objective of this study is a numerical study, so the authors should have investigated each segment of the force-displacement curves and provided more evidence for the observed errors. Moreover, the force-displacement curve is very sensitive to the dimensions of the specimens and their standard deviation. How did the authors minimize this error?
The paper lacks a deep scientific analysis. The simulation validation was adequately performed, but no further investigation was done. Adding a section for a detailed sensitivity analysis on the simulation parameters is suggested.
Author Response
Comment 1:Please describe the mesh dependency method in detail. Which parameter was considered for the mesh size analysis?
Response 1: The finite element mesh is locally refined at the notch following the mesh size refinement condition proposed with the relation between length scale parameter and mesh size. The mesh refinement condition is given in the revised submission. Mesh sensitivity is performed but details are not given since many articles are available on mesh sensitivity study. I can report these details in my submission, if the cited articles on mesh study are not sufficient.
Comment 2: The root-mean-square (RMS) of the error should be calculated for the given results in Fig. 3, and the authors should discuss the origin of the errors.
Response 2: In the present work, the standard deviation is calculated between simulation and mean test data in phase-field damage results. Mean test data is calculated from 3 tests performed for each climatic condition. Therefore, we consider standard deviation presented in the submission would be sufficient to validate the results.
Comment 3: The main objective of this study is a numerical study, so the authors should have investigated each segment of the force-displacement curves and provided more evidence for the observed errors. Moreover, the force-displacement curve is very sensitive to the dimensions of the specimens and their standard deviation. How did the authors minimize this error?
Response 3: Angular sample is clamped approximately at 22 mm on both free ends and the clamped volume is assumed to be rigid. Therefore, the clamped volume of material is not considered in the numerical investigation. We have considered the finite element model analogous to the test setup by applying displacement boundary condition at approximately at 22 mm cut section on both free ends. Schematic figures are added to the submission
Comment 4: The paper lacks a deep scientific analysis. The simulation validation was adequately performed, but no further investigation was done. Adding a section for a detailed sensitivity analysis on the simulation parameters is suggested.
Response 4: Tensile test investigations performed for different strains rates are provided to show sensitivity of the viscoelastic parameters.
Reviewer 3 Report
Comments and Suggestions for Authors
Dear Authors,
I have completed the review of your manuscript titled "Numerical investigation of fracture behaviour of polyurethane adhesives under the influence of moisture". However, I would like to recommend some revisions before further consideration.
1) Please provide more details about the experimental procedure used to prepare the aged samples under different relative humidity conditions. Were there any challenges or limitations in maintaining the desired humidity levels?
2) In the finite element implementation, how was convergence ensured, especially considering the non-linear nature of the problem?
3) The identified viscoelastic parameters for the polyurethane adhesive are based on curve fitting using the Matlab optimization toolbox. Please provide more details about the optimization process and the criteria used for selecting the best-fitting parameters. Were there any sensitivity analyses conducted to assess the robustness of the identified parameters?
4) The critical energy release rate (Ec) is an important parameter in the phase-field damage model. How sensitive is the crack propagation behavior to variations in Ec? Were there any discussions or analyses conducted to investigate the influence of Ec on the simulation results?
5) The comparison between the experimental and simulation data is shown in Figure 3, indicating a good agreement. Please provide additional statistical measures of the comparison, such as the coefficient of determination (R-squared value) or root mean square error (RMSE), to support the validation of the identified material parameters?
6) You mention that the phase-field damage model is a promising approach for describing crack evolution. Please discuss any limitations or challenges associated with the phase-field model in the context of this study. Are there any potential areas of improvement or future research directions that you would recommend?
7) Based on the results and conclusions of your study, what are the key takeaways and contributions to the existing body of knowledge in the field of viscoelastic behavior and damage analysis of polyurethane adhesives? How does this work advance the understanding of the material behavior and provide valuable insights for future research endeavors?
Author Response
Comment 1: Please provide more details about the experimental procedure used to prepare the aged samples under different relative humidity conditions. Were there any challenges or limitations in maintaining the desired humidity levels?
Response 1: Experimental investigation performed on the commercial polyurethane sample under IGF-project 19730 N and cited in the reference 66. I have used the experimental results performed by project partners to develop material model for numerical investigations.
Comment 2: In the finite element implementation, how was convergence ensured, especially considering the non-linear nature of the problem?
Response 2: The problem is solve in many time steps (atleast 100) using Newton’s iterative method. The Newton’s iterative time step is solved in several pseudo-time steps with a tolerance value of 10^-8.
Comment 3: The identified viscoelastic parameters for the polyurethane adhesive are based on curve fitting using the Matlab optimization toolbox. Please provide more details about the optimization process and the criteria used for selecting the best-fitting parameters. Were there any sensitivity analyses conducted to assess the robustness of the identified parameters?
Response 3: Nelder and Mead optimization algorithm is used in the curve fitting process. The details are given in revised version of the submission.
Comment 4: The critical energy release rate (Ec) is an important parameter in the phase-field damage model. How sensitive is the crack propagation behavior to variations in Ec? Were there any discussions or analyses conducted to investigate the influence of Ec on the simulation results?
Response 4: Critical energy release rate Ec have significant effect on the force-displacement curve. With the smaller Ec the maximum force required for sample to fail increase with smaller value the sample fails early. The critical energy release rate reported in the submission corresponds to the property of the specific material. Several publications are reported to discuss the sensitivity of Ec, therefore we have not reported these studies in our investigations.
Comment 5: The comparison between the experimental and simulation data is shown in Figure 3, indicating a good agreement. Please provide additional statistical measures of the comparison, such as the coefficient of determination (R-squared value) or root mean square error (RMSE), to support the validation of the identified material parameters?
Response 5: In the present work, the standard deviation is calculated between simulation and mean test data in phase-field damage results. Mean test data is calculated from 3 tests performed for each climatic condition. Therefore, we consider standard deviation presented in the submission would be sufficient to validate the results. Upon request I can also provide the results of the simulation and test series comparison performed for different humid conditions.
Comment 6: You mention that the phase-field damage model is a promising approach for describing crack evolution. Please discuss any limitations or challenges associated with the phase-field model in the context of this study. Are there any potential areas of improvement or future research directions that you would recommend?
Response 6: Need for the higher order phase-field damage model is mentioned in the section conclusion and future work
Comment 7: Based on the results and conclusions of your study, what are the key takeaways and contributions to the existing body of knowledge in the field of viscoelastic behavior and damage analysis of polyurethane adhesives? How does this work advance the understanding of the material behavior and provide valuable insights for future research endeavors?
Response 7: The primary motive of the work is to investigate the moisture influence on the fracture behaviour. The ductility of the material is significantly effected by the diffusion of moisture. Further, need for micro-mechanical model based on statistical polymer chain mechanics is discussed in the submitted revision.
Reviewer 4 Report
Comments and Suggestions for Authors
SUMMARY
The article presented for review is relevant for modern polymer science and engineering developments. Numerical studies of the destruction of polyurethane adhesives under the influence of moisture have been performed. The relevance of the study is due to the fact that polyurethane adhesives are widely used in the production of lightweight components. These polyurethane adhesives are hygroscopic in nature, which results in deterioration of the material's mechanical properties due to aging in environmental conditions. The authors' work is devoted to studying the effects of nitrogen on exposure to harmful substances. The test results were used to determine the parameters of material damage in a phase field based on a numerical study.
The reviewer of the article is suitable for the journal “Polymers” in its subject matter. The authors have done a lot of work and gained some important knowledge. Their developments will be useful for readers of the magazine, as well as for engineering workers. However, the reviewer believes that the article in its current form is not yet ready for publication. She needs massive changes. The reviewer's comments are listed below.
COMMENTS
1. The abstract submitted by the authors does not meet the requirements of the journal. It needs to be reworked. The scientific novelty of the research should be stated at the beginning of the abstract. What scientific gaps exist in studying the degradation of polyurethane adhesives under the influence of moisture? This needs to be indicated.
2. The abstract does not contain a formulation of an applied engineering problem. The abstract describes only the relevance of the topic of polyurethane adhesives. It is necessary to show what specific problem or barrier exists in the study and application of polyurethane adhesives in specific conditions of exposure to moisture. What do engineers and materials scientists face?
3. The end of the annotation informs that tests were carried out and the tensile strength of samples exposed to various humidity conditions was determined. However, no quantitative expressions of the results obtained are provided. They need to be added, we need specific numbers.
4. It is recommended to increase the number of keywords to 6-7. This will help interested authors and readers find this publication faster in search algorithms.
5. The “Introduction” section is represented by 28 literature sources. This is not enough to talk about a complete analysis of the current state of the issue. It is proposed to increase the list of analyzed works in the “Introduction” section to 35-40 titles, then the current state of the issue will be more fully considered.
6. It is also necessary to show in the “Introduction” section the problems of more applied problems and more fundamental ones. That is, a review of those works that studied the applied applications of polyurethane adhesives must be separated from a review of works that dealt with fundamental problems of studying the composition, structure and properties of polyurethane adhesives in various environments and influences.
7. The second section looks uninformative. It should be renamed to the Methods subsection and merged with section number 3.
8. Section 4 is probably also desirable to combine with sections 2 and 3 and denote them under the general title “Materials and Methods”.
9. Numerical studies in section 5 should be renamed to the “Results” section.
10. It is necessary to add a “Discussion” section in which to discuss in detail the results obtained and compare them with the results of authors who dealt with similar problems.
11. It is methodologically incorrect to end the penultimate section with Figure 4. A smoother transition from discussion to conclusions should be added.
12. Conclusions need to be specified and Section 6 Summary should be renamed to the Conclusions section.
13. It is necessary to present in the Conclusions section the scientific novelty, practical significance, recommendations for the real construction industry, as well as prospects for the development of this research in the future.
14. References to literature should be removed from the Conclusions section. Currently, line 246 provides a link to literature source No. 53. You should already talk about your own scientific conclusions in the “Conclusions” section, and move all links to the previous Discussion section.
15. The bibliography, which includes 59 titles, contains a large amount of literature older than the last 10, 50 and even 100 years. For example, literature source No. 47 dates back to 1921. Still, for world-class research, more modern sources should also be analyzed. Therefore, it is recommended to add 15-20 works for the period 2019-2024 so that the current state of the issue is reflected more fully.
Once all changes have been made, the reviewer would like to take a second look at the paper to make sure it is suitable for publication in Polymers. The general conclusion is Major Revisions.
Author Response
Comment 1: The abstract submitted by the authors does not meet the requirements of the journal. It needs to be reworked. The scientific novelty of the research should be stated at the beginning of the abstract. What scientific gaps exist in studying the degradation of polyurethane adhesives under the influence of moisture? This needs to be indicated.
Response 1: Abstract is modified in the revision to provide summary on the work performed in the submission
Comment 2: The abstract does not contain a formulation of an applied engineering problem. The abstract describes only the relevance of the topic of polyurethane adhesives. It is necessary to show what specific problem or barrier exists in the study and application of polyurethane adhesives in specific conditions of exposure to moisture. What do engineers and materials scientists face?
Response 2: The abstract is improved as per suggested comments. This also provides information to use the work involved to numerically investigate the mechanical behaviour of the viscoelstic polymers
Comment 3: The end of the annotation informs that tests were carried out and the tensile strength of samples exposed to various humidity conditions was determined. However, no quantitative expressions of the results obtained are provided. They need to be added, we need specific numbers.
Response 3: Results of the tensile test are reported in the revision investigation for different strain rates
Comment 4: It is recommended to increase the number of keywords to 6-7. This will help interested authors and readers find this publication faster in search algorithms.
Response 4: Increased to 6 keywords
Comment 5: The “Introduction” section is represented by 28 literature sources. This is not enough to talk about a complete analysis of the current state of the issue. It is proposed to increase the list of analyzed works in the “Introduction” section to 35-40 titles, then the current state of the issue will be more fully considered.
Response 5: More relevant literature is cited in the introduction
Comment 6: It is also necessary to show in the “Introduction” section the problems of more applied problems and more fundamental ones. That is, a review of those works that studied the applied applications of polyurethane adhesives must be separated from a review of works that dealt with fundamental problems of studying the composition, structure and properties of polyurethane adhesives in various environments and influences.
Response 6: So far very the literature available on the crosslinked polyurethane adhesive are limited and not relevant to the scope of the research on “fracture behaviour of samples aged under the influence of moisture”
Comment 7: The second section looks uninformative. It should be renamed to the Methods subsection and merged with section number 3.
Response 7: Section 2 and 3 are merged under section title Material model formulation. This section discusses the kinematic of finite-strain theory and essential free energy functions chosen to derive constitutive equation and respective governing equations.
Comment 8: Section 4 is probably also desirable to combine with sections 2 and 3 and denote them under the general title “Materials and Methods”.
Response 8: Section 4 is now section 3 titled Governing balance equations. This section deals with the finite element implementation detailing the weak forms of the governing equations. Further, the linearized components of the tangent tensor is given along with the solution components to solve with Newton’s method. I believe, combining with earlier section might be difficult for the reader who want to understand implementation. Therefore, we prefer to have separate sections for the reader to toggle to the required sections.
Comment 9: Numerical studies in section 5 should be renamed to the “Results” section.
Response 9: Renamed the section
Comment 10: It is necessary to add a “Discussion” section in which to discuss in detail the results obtained and compare them with the results of authors who dealt with similar problems.
Response 10: The detailed on the results have been discussed in the results sections. Further the short coming in the work is discussed on the section “conclusion and futur work”
Comment 11: It is methodologically incorrect to end the penultimate section with Figure 4. A smoother transition from discussion to conclusions should be added.
Response 11: Conclusion and future work section is added to discuss on the requirements for the further improvement of the material model and the prospects for the future work to improve material model to better describe the crosslinked polymers
Comment 12: Conclusions need to be specified and Section 6 Summary should be renamed to the Conclusions section.
Response 12: Renamed summary title to conclusion and future work
Comment 13: It is necessary to present in the Conclusions section the scientific novelty, practical significance, recommendations for the real construction industry, as well as prospects for the development of this research in the future.
Response 13: Answered in the queries 11-12
Comment 14: References to literature should be removed from the Conclusions section. Currently, line 246 provides a link to literature source No. 53. You should already talk about your own scientific conclusions in the “Conclusions” section, and move all links to the previous Discussion section.
Response 14: I have modified the section as suggested. Answers to queries 11-12 are also relevant to this query
Comment 15: The bibliography, which includes 59 titles, contains a large amount of literature older than the last 10, 50 and even 100 years. For example, literature source No. 47 dates back to 1921. Still, for world-class research, more modern sources should also be analyzed. Therefore, it is recommended to add 15-20 works for the period 2019-2024 so that the current state of the issue is reflected more fully.
Response 15: The sources cited are relevant to the discussion as these are the base formulation needed in the development of the work.
Round 2
Reviewer 1 Report
Comments and Suggestions for Authors
The manuscript “Numerical investigation of fracture behaviour of polyurethane adhesives under the influence of moisture” describes a finite-strain viscoelastic material model coupled with the phase-field model to investigate the fracture behaviour of these adhesives in different moisture.
The results were compared, and the conclusions were made.
The manuscript is original and has merit, but it needs to be improved.
The chemical properties of the adhesive need to be presented.
What is the chemical structure of the adhesive?
Does the adhesive need a curing agent to harden?
it is necessary to write more text on the polyurethane adhesive. Taking this study into account, where will it be used?
The study did not consider the crystallization process, shrinkage process or swelling process in the model. The adhesive behaviour can be influenced by be all these process when subjected to the temperature and the moisture conditions.
The authors must compare the results with the similar studies found in the literature.
Use all references that adopt written English. Chanege the following references and similar ones.
65. Zimmer, B. Hygrothermale Alterung amorpher Polyurethannetzwerke; 2021. https://doi.org/http://dx.doi.org/10.22028/D291-36700. 546
66. Wulf, A.; Hesebeck, O.; Koschek, K.; M. Brede, Mayer, B.; Josyula, S.; Diebels, S.; Zimmer, B.; Possart, W. Berechnung des instationären mechanischen Verhaltens von alternden Klebverbindungen unter Einfluss von Wasser auf den Klebstoff. DVS 548 Forschungsvereinigung 2021, 469, 0–266
Author Response
Comment 1: The chemical properties of the adhesive need to be presented. What is the chemical structure of the adhesive? Does the adhesive need a curing agent to harden? it is necessary to write more text on the polyurethane adhesive. Taking this study into account, where will it be used?
Response 1: The chemical structure of basic crosslinked polyurethane adhesive is manufactured by mixing two polyether alcohol components (resin) and an isocyanate component (hardener). The general chemical structure of the basic polyurethane adhesive PU9010 is described in detail in the references [2] and [66]. The current work is conducted to investigate on the commercially polyurethane adhesive manufactured by Dupont under the trade name Betaforce 2850L. Unfortunately, we do not have information on the components of the Betaforce 2850L, therefore we can provide this information. More details on the polyurethane adhesive and their commercial application in the industry are given in the current form of the submission.
Comment 2: The study did not consider the crystallization process, shrinkage process or swelling process in the model. The adhesive behaviour can be influenced by be all these process when subjected to the temperature and the moisture conditions.
Response 2: Swelling in the polymer is considered in the material modelling. Unfortunately, we did not make experimental investigations to understand the swelling phenomenon due to moisture diffusion. IR Spectroscopy investigation no chemical interaction between polyurethane adhesive and water was observed. The influence of the moisture diffusion on the glass transition temperature and their effects on the crystallization process is the scope of our future work. Unfortunately, I can not address these question in the present form of the submission.
Comment 3: Use all references that adopt written English. Change the following references and similar ones.
- Zimmer, B. Hygrothermale Alterung amorpher Polyurethannetzwerke; 2021. https://doi.org/http://dx.doi.org/10.22028/D291-36700. 546
- Wulf, A.; Hesebeck, O.; Koschek, K.; M. Brede, Mayer, B.; Josyula, S.; Diebels, S.; Zimmer, B.; Possart, W. Berechnung des instationären mechanischen Verhaltens von alternden Klebverbindungen unter Einfluss von Wasser auf den Klebstoff. DVS 548 Forschungsvereinigung 2021, 469, 0–266
Response 3: Reference [65] is replaced with reference [66] in the current form. Unfortunately, [65] in its current form [68] can not be replaced as we have not published the experimental investigation in English. In the current form, the test results plotted in the current form to compare with the FE results are the same results discussed in the reference [68].
The updates to answer the comments on the earlier submission are highlighted in blue color in current form of the submission.
Reviewer 2 Report
Comments and Suggestions for Authors
First of all thanks to the authors for their efforts in answering the comments. Although the authors have attempted to revise the paper based on the raised comments, some issues remain and must be addressed.
1- The authors have overlooked the first comment.
2- The Root Mean Square (RMS) of the simulated and experimental curves must be calculated to compare the predicted and experimental results. The standard deviation has been used to evaluate the performed experiment, while RMS should be employed to compare experimental and finite element (FE) results.
3- Clearly specify which mechanical parameters were considered for the mesh sensitivity analysis and include the mesh sensitivity graph that illustrates how the optimum number of elements was determined accordingly.
After addressing the provided comments, the paper could be considered for publication in Polymers.
Author Response
Comment 1- Please describe the mesh dependency method in detail. Which parameter was considered for the mesh size analysis?
Response 1- The stress concentration is higher at the notch; therefore, the finite element mesh is locally refined at the notch. Viscoelastic parameters of the dry sample listed in the submission is used in the mesh sensitivity study. Mesh sensitivity study is investigated for different mesh size for a length scale parameter of 9.13 mm.
Comment 2- The Root Mean Square (RMS) of the simulated and experimental curves must be calculated to compare the predicted and experimental results. The standard deviation has been used to evaluate the performed experiment, while RMS should be employed to compare experimental and finite element (FE) results.
Response 2- We have always compared FE and test results with standard deviation. As per your suggestion, I have adopted RMS error to compare the FE and test results in updated submission.
Comment 3- Clearly specify which mechanical parameters were considered for the mesh sensitivity analysis and include the mesh sensitivity graph that illustrates how the optimum number of elements was determined accordingly.
Response 3- I have introduced documentation on the mesh sensitivity study to understand the optimal mesh size. Mesh size is dependent primarily on the length scale parameter in the phase-field damage modelling. The initial crack introduced in the test specimen is assumed as the length scale parameter in the current investigation. In the current form of the submission, I have not investigated different length scale parameters, as there are many publications dedicated to studying the length scale parameter on damage propagation in phase-field damage modelling.
The updates to answer the comments on the earlier submission are highlighted in blue color in current form of the submission.
Reviewer 4 Report
Comments and Suggestions for Authors
The authors have revised the article, it has improved significantly. The comments have been answered, the main comments have been corrected.
I have no more comments, the article can be accepted in its current form.
Author Response
Thank you for accepting the submission in current form
Round 3
Reviewer 2 Report
Comments and Suggestions for Authors
-